# Effectiveness of Different Application Parameters of Neuromuscular Electrical Stimulation for the Treatment of Dysphagia after a Stroke: A Systematic Review

**DOI:** 10.3390/jcm9082618

**Published:** 2020-08-12

**Authors:** Isabel Diéguez-Pérez, Raquel Leirós-Rodríguez

**Affiliations:** Faculty of Physical Therapy, University of Vigo, Campus a Xunqueira, s/n., 36005 Pontevedra, Spain; isadieper@gmail.com

**Keywords:** deglutition, stroke, electric stimulation, neurology, physical therapy modalities

## Abstract

Background: Dysphagia causes severe complications among people with a stroke. Physiotherapy allows the cure of this pathology, and among the tools it offers is neuromuscular electrical stimulation. However, this is a technique that has not been protocolized. Therefore, it was considered necessary to carry out a systematic review on the efficacy of the various parameters of application of the neuromuscular electrical stimulation in dysphagia generated after a stroke. Methods: A systematic search for publications was conducted in March 2020 in the Pubmed, Cinahl, Medline, Web of Science and Scopus databases, using as search terms: Electric stimulation therapy, Deglutition disorders and Stroke. Results: 21 articles were obtained in which the application of neuromuscular electrical stimulation was applied in isolation (*n* = 7) or in combination with other techniques such as strengthening exercises and manual therapy techniques (*n* = 14), with this second modality of treatment having greater benefits for patients. Conclusion: The greatest efficacy of this technique is reached when applied at 60-80 Hz, 700 μs of pulse duration, at the motor intensity threshold and in sessions of 20–30 min.

## 1. Introduction

Dysphagia (DP) is a condition characterised by the loss of swallowing function [1], due to the alteration of the oral route and of the pharyngeal and esopharyngeal phases [2], hindering or impeding the passage of the alimentary bolus from the mouth to the stomach [3,4]. It can be caused by different disorders, such as affectations of the central nervous system (e.g., strokes or Parkinson’s disease), structural alterations (as a consequence of cranial surgical interventions or laryngectomies) [5] or motor disorders (which block the coordination of muscular actions) [3,4].

DP affects 13% of people over 65 years of age and approximately 33% of people over 80 years of age [6], although it has a prevalence of 55% after a stroke [7]. The most common complications associated with this disorder are: suffocation, aspiration pneumonia, malnutrition and decreased physical activity; these must be addressed with special care and individually [8,9,10]. In turn, these complications are frequently associated with weight loss, low body mass index and aggravation of age-related sarcopenia [11]. All this leads to effects on the quality of life and the decreased survival of DP patients, associated with an increase in the socio-economic (due to the frequent need of hospitalisation) [12], human (due to the need of the patients to be attended to by carers) [13] and psychological burden (due to the emotional load and decreased independence of the patients) [13].

Therefore, it is necessary to establish an effective and efficient treatment for DP patients. There is a wide range of possible treatment options, including: pharmacological treatment (controversial, thus it is currently not in use) [14], compensatory techniques (provide improvements, although without completely resolving the alteration of swallowing) and physiotherapeutic methods (important in the treatment of swallowing disorders because they have an impact not only on eating restoration, but also on other orofacial functions such as facial expression and speech [15,16]. The fundamental aim of the orofacial physiotherapeutic approach is to strengthen the swallowing musculature (facial, suprahyoid and infrahyoid) to restore its tone, power, movement amplitude, speed and coordination through indirect techniques (or “empty”, with the mouth open) or direct techniques (using maneuvers performed with foods or liquids to facilitate the training in conditions similar to those of daily living) [17]. At the same time, in recent years the application of electrical stimulation has been incorporated through different application modalities: transcranial, invasive (pharyngeal stimulation), or surface stimulation (neuromuscular). Transcranial direct current stimulation promotes brain plasticity by tonic stimulation with weak direct currents [18]. Electrical pharyngeal stimulation is one neurostimulation technique that has been shown to promote plasticity in healthy individuals and achieve measurable improvements in swallowing function [19,20]. However, currently, these two modalities are less standardized and, although the results of the investigations carried out to date are promising, there are few clinical trials to support them [21,22,23].

Finally, the surface or neuromuscular electrical stimulation (NMES), due to its capacity to increase muscular performance, is a technique to be taken into account to improve the efficacy of the physiotherapeutic treatment in DP patients [24]. The NMES facilitates a muscle contraction and is used on innervated muscle to recruit motor units and increase muscle strength [25]. However, its appropriate application parameters have not yet been described, such as intensity (sensitive threshold, motor or a specific range), frequency, impulse time and electrode placement [25,26]. Therefore, it was considered necessary to carry out a literature review of the scientific research published in recent years, with the aim of evaluating the efficacy of the various parameters of application of the NMES in DP generated after a stroke.

## 2. Methods

The PICO question was then chosen as follows: P—population: DP patients after stroke; I—intervention: NMES; C—control: conventional physiotherapy techniques; O—outcome: swallowing efficacy (level of oral intake presence and severity of penetration and aspiration, oral and pharyngeal transit times…); S-study designs: experimental studies. The systematic search of publications was conducted in March 2020 in the Pubmed, Cinahl, Medline, Web of Science and Scopus databases, using the following words as search terms of the Medical Subject Headings (MeSH) thesaurus: *Electrical stimulation therapy, Deglutition disorders* and *Stroke*; and *Dysphagia* as a free term. The search strategy according to the focused PICOS question is presented in Table 1.

For the selection of results, the inclusion criteria established that the articles must have been published in the last five years, that the sample of studies consisted exclusively of patients with DP after a stroke and that the authors applied a treatment intervention that included non-invasive NMES. On the other hand, studies were excluded from this review if they had a non-experimental methodology, their full text was not available, and if they applied NMES transcranially. The search and selection process are shown in detail in Figure 1.

After screening the data, extracting, obtaining and screening the titles and abstracts for inclusion criteria, the selected abstracts were obtained in full texts. Titles and abstracts lacking sufficient information regarding inclusion criteria were also obtained as full texts. Full text articles were selected in case of compliance with inclusion criteria by the two reviewers using a data extraction form. The two authors independently collected the following data from the included articles for further analysis: demographic information (title, authors, journal and year), study-specific parameter (study type, number of treated patients, duration of intervention, number of sessions, techniques of physical therapy included in the intervention, follow-up and drop-out), NMES application parameters (frequency, impulse time, intensity, electrode location and stimulation device). Furthermore, the Jadad scale was used to assess the high quality of studies.

## 3. Results

A total of 21 articles were found, of which eleven were focused on experimental researchs [27,28,29,30,31,32,33,34,35,36,37] and nine were randomized-controlled trials [38,39,40,41,42,43,44,45,46,47]. The methodological characteristics of the analysed studies are shown in Table 2 and, in Table 3, the characteristics of the interventions applied in them are presented. 

The application of NMES was not conducted using the same parameters in the analysed investigations. Regarding the application frequency, the choice of 80 Hz was the most widely used [27,28,29,30,31,32,33,35,36,37,39,40,41,42,45], although, in several cases, this parameter was not mentioned [34,38,44,46] and in another two studies the authors selected lower frequencies (60 [47] and 30 Hz [43]). In all cases, the results were statistically significant; even those studies which used lower frequencies obtained significant reductions in oral and pharyngeal transit time [47], significant decreases in the severity of penetration and aspiration [43] and significant increases in the level of oral intake and quality of life [43].

The impulse time was also a parameter that varied depending on the study analysed. The most frequent duration was 700 μs [28,29,30,32,33,35], although there were studies that used shorter (300 μs [31,40,41]) and longer durations (300 [27,36,37], 700 [39] and 800 [44] ms). In all the analysed cases, the results were significantly positive; even the studies that used shorter impulse times detected improvements in the quality of sleep, level of oral intake, presence and severity of penetration and aspiration, and capacity to communicate and swallow (higher elevation of the pharynx, closing of the epiglottis and coaptation of the pharyngeal wall), although more notably in those cases in which NMES was combined with Mendelsohn maneuvers (and other conventional DP treatment techniques) compared to its use as a single treatment technique [31,40,41].

Regarding the intensity of the application, most of the obtained studies were aimed at reaching the motor threshold as the patient’s tolerance to the current increased with the sessions [28,29,30,35,42,43,45,46,47]. In contrast, other studies reached the pain threshold of the patient [31] or aimed to reach a certain level of intensity as the patient’s tolerance increased with each session: up to 7 [37,38], 8.5 [32], 12.5 [41], 20 [36], 25 [40] and 28 mA [37]. Specifically, Park et al. [35] compared the effects of applying NMES at different intensities: an intervention group received it at motor threshold, whereas the control group received it at sensitivity threshold. Their results showed that the swallowing function, presence and severity of penetrations and aspirations and the mobility of the hyoid bone improved significantly in the patients who received the intervention at higher intensity.

The electrodes were preferentially placed in the anterior side of the neck: suprahyoid [27,31,32,33,36,37,38,39,42,43,44,45,46,47] or infrahyoid area [35], thyroid muscles [28,41,44], and lateral regions of the pharynx [40], although some authors also applied NMES in the cranium: in the paretic facial area [29], orbicularis orbis muscle [30] and masseters [37]. Two of the analysed articles were focused on comparing the effects of different application areas. Lee et al. [37] compared the application in the masseter and suprahyoid muscles with the stimulation of the suprahyoid muscles alone, and the results of the American Speech-Language-Hearing Association National Outcome Measurement System (ASHA-NOMS), the Functional Dysphagia Scale and the Penetration Aspiration Scale did not show significant differences between groups in any case. On the other hand, Meng et al. [42] compared the stimulation at both sides of the suprahyoid muscles and laterally at the thyroid cartilage with the application of NMES on the genohyoid and mylohyoid muscles. In this case, the results of the Water-Swallow Test, Repetitive Saliva Swallowing Test, Dysphagia Outcome and Severity Scale and the Videofluoroscopic Dysphagia Scale (VDS) indicated that the changes were greater with the application of NMES on the genohyoid and mylohyoid muscles. Regarding the other two studies that used cranial application points, Choi et al. [29], with the stimulation of the paretic facial region, and Oh et al. [30], with the stimulation of the orbicularis orbis muscle, obtained significant improvements in swallowing function and maximum strength of the cheeks and lips.

The duration of the treatment sessions ranged between 20 [27,33,37,44,47] and 60 min [34,38,46], although the most frequent duration was 30 min [28,29,30,31,32,35,36,41,42,43]. In all cases, the results were significantly positive; however, it is important to highlight that those interventions with longer sessions did not obtain improvements in pharyngeal closure [47] or in the Brunnstrom’s motor stages [43] and, although the analysed patients did improve in the intensity of perceived pain, level of oral intake, swallowing, aspiration and penetration events and oral and pharyngeal transit times, such improvements were greater in the groups of patients who received traditional DP therapy [47] or McNeill’s therapy [46] with respect to those who received NMES.

In most cases, the interventions were one month long [27,28,29,30,32,33,34,38,43], whereas one of the interventions was only 10 days long [37] and another intervention was two months long [41]. The application frequency also ranged, from two sessions per day [37] to three sessions per week [33,40], with the most usual schedule being five sessions per week [27,28,29,30,32,34,35,36,38,41,42,43,47]. Byeon et al. [31], Hamada et al. [39] and Guillén-Sòla et al. [45] did not define the frequency of the sessions in their interventions; they only detected significant improvements in the development of pulmonary infections [39].

The NMES application device was, in most cases, VitalStim^®^ (Chatanooga, United Kingdom) [27,28,29,30,31,35,36,45,46] and, in the study in which this aspect was specified, Ampcare-ESP^®^ (Ampcare, USA) [43].

Regarding the applied interventions, seven studies used NMES as a single treatment technique [27,28,29,30,33,35,36]. Particularly, Park et al. [28] and Mituuti et al. [33], after their interventions, detected the effectiveness of NMES to improve the quality of life of the participants [33], the penetration and aspiration events and swallowing (both in the oral phase and in the pharyngeal phase in general), although their specific parameters (lip closure, formation of the bolus, chewing efficacy, apraxia, contact of the tongue with the palate, premature loss of the bolus, activation of the uvula, presence of residues, and higher pharynx and oral and pharyngeal transit times) did not reach significant changes [28]. Byeon [27] and Byeon and Koh [36] compared the efficacy of NMES with that of the Masako maneuver and with the stimulation of the anterior pillar of the palate (or palatoglossal arch), respectively. Their results showed that swallowing (evaluated using the Functional Dysphagia Scale) improved with all interventions (without statistical differences between groups) [27,36].

The rest of the articles combined the application of NMES with another physiotherapeutic technique. In the most frequent combination, the treatment sessions included one or several techniques of general or conventional therapy for DP, for example: thermal stimulation techniques [34,40,41], tongue-strengthening exercises [34,37,40,41], oro-facial and pharyngeal musculature and mouth closure strengthening exercises [32,34,37,41,42,43,45], exercises to raise the larynx and favour the closing of the vocal cords [41,42], Shaker exercises [41,42], Masako [41] and Mendelsohn maneuvers [40,41,42], McNeill therapy [46], craniocervical postural correction [34,38,40,42], respiratory pattern correction [34,42,45] and modifications of the dietary habits [34,38,40,42]. In all these cases, the sample studies improved both when NMES or the traditional therapy was applied alone and in combination, although the combined application obtained better improvements in all cases.

Particularly, Konecny and Elfmark [47] applied, in combination with NMES, an oro-facial rehabilitation programme that included: postural correction and respiratory rehabilitation therapy for the laryngeal closure, thermal stimulation, strengthening exercises for the tongue and facial and labial musculature, and swallowing training (by pressing the palate with the tongue and strengthening the bite and the closure of the vocal cords). Their results showed that both treatment methods, separately, reduced the oral and pharyngeal transit times, and that the group that received the two therapies combined obtained significantly better results [47]. Lastly, Zeng et al. [44] included in their treatment protocol, in addition to NMES, the conventional pharmacological therapy (platelet inhibitors, lipid-lowering agents, antihypertensive drugs, euglycemic drugs, free radical scavengers and/or facilitators of microcirculation) and empty or dry swallowing training: slight massage in the cheeks, tongue, retropharyngeal wall, palatopharyngeal area and lips with a cotton pad dipped in ice water. In this case, the swallowing function and the levels of anxiety, cognitive disorder and psychomotor involvement also improved significantly with both therapies, separately, and the group that received the two therapies combined obtained even better results.

## 4. Discussion

The aim of the present study was to establish the efficacy of the various parameters of application of the NMES in DP generated after a stroke. The analysis of the literature on this specific topic shows that this technique improves the resolution of the clinical manifestations of this pathology. The intensity of the muscular contraction caused by NMES is controlled by manipulating the parameters of frequency, intensity and duration of the impulse [48,49].

Frequencies below 40–50 Hz induce the recruitment of more slow-twitch fibres (type I), which are more resistant to fatigue [50] (as can be observed in the study of Sproson et al. [43]), whereas higher impulse frequencies recruit more fast-twitch fibres (type II), which are less resistant to fatigue [50]. However, it is important to take into account that the muscle fibre recruitment pattern of NMES is different to the physiological recruitment, preferentially favouring the activation of fast motor units over that of slow motor units [51], which is beneficial for the treatment of DP, since the swallowing muscles are a predominant component of fast fibres [52].

To provoke an adequate muscle contraction, in addition to employing a stimulation frequency of 50–100 Hz [53], it is recommended to apply the highest intensity possible [54]. Carnaby-Mann and Crary [25] suggested that the key to optimising the effectiveness of NMES lies in achieving the maximum muscular tension during the application, which depends on the maximum evoked and voluntary forces. To reach it adequately, it is essential to manipulate with precision the parameters of intensity and frequency, which, if increased progressively, can produce more vigorous contractions [55,56,57]. However, intensity also influences the comfort of the patient (higher intensities are usually worse tolerated). Therefore, there must be an adequate combination between frequency and intensity to reach a quality muscular contraction. The duration of the impulse must also be taken into account; in fact, Grill and Mortimer [58] reported that, the shorter this is, the greater the intensity of the stimulus must be to obtain a muscle response. That is, the duration of the impulse is inversely proportional to the specificity of the applied stimulation.

Most of the analysed studies used an impulse frequency in the range of 60 [47] to 80 Hz [27,28,29,30,31,32,33,35,36,37,39,40,41,42,45], thereby stimulating mainly type II fibres (adjusting to the needs of the target muscles) and obtaining significant results in all cases, except in Mituuti et al. [33], Lee et al. [37], Hamada et al. [39] and Guillén-Sòla et al. [45], probably due to the fact that, in the first three studies, the impulse duration used was longer than in the rest of the studies and, in the last study, the duration of the session was 40 min, which could have caused a counterproductive fatigue as a result of the excessive length of the treatment [50]. It is important to highlight that the study which used a frequency of 60 Hz [47] had an intervention duration that was much shorter than that of the rest of the studies, obtaining similar results with respect to the rest of the studies. Similarly, despite the choice of a low stimulation frequency, Sproson et al. [43] also obtained improvements that could be due to the day-to-day increase in the impulse duration.

The intensities used were very varied, although in most cases they reached the motor threshold [28,29,30,31,32,35,42,43,45,46,47]. Mituuti et al. [33] used the sensitivity threshold without obtaining significant improvements, and Park et al. [35] detected differences between the motor threshold group and sensitivity threshold group, with the one receiving NMES up to the motor threshold obtaining better results.

Most of the analysed investigations applied NMES with short impulses (700 µs) [28,29,30,31,32,33,35,40,41]. However, some studies used longer impulses (between 300 [27] and 800 ms [44]), which also obtained benefits; this could be due to the daily application of the treatment or to what has been suggested by Sarafoleanu [59], who stated that, with low-intensity and long impulses, it is possible to increase the recruitment of the motor units.

Some of the analysed studies requested simultaneous contraction while the NMES was applied [35,36,43,45]. Mituuti et al. [33] was the only study that did not obtain significant effects, although this phenomenon could be due to the fact that it was also the only one in which the treatment was applied up to the sensitivity threshold. It is important to take into account that this is a controversial aspect of the application of NMES, since its administration, combined with simultaneous voluntary contraction, has been previously tested by Vanderthommen and Duchateau [53], who concluded that these two methods should not be used together due to the high metabolic demand caused by such a combination and to the possible fatigue that it could cause in the muscle fibres. On the other hand, Sarafoleanu and Enache [59] concluded that the combination of these two treatments induced greater muscular adaptations. The results obtained in this review are in line with the conclusions of the latter authors. Furthermore, the studies that applied NMES passively [27,28,29,30,31,32,34,37,38,39,40,41,42,44,46,47] also obtained significant improvements, except in the studies of Byeon [27] and Lee et al. [37], in which the impulse duration was 300 ms and the sessions were 20 min long, which may have been insufficient for an effective dosage of the treatment.

The adequate placing of the electrodes is a fundamental aspect to reach the maximum efficacy of NMES. In most of the analysed studies, the electrodes were placed in the anterior side of the neck [27,28,31,32,33,35,36,37,38,39,40,41,42,43,44,46,47]. Other authors placed the electrodes in facial areas (paretic [29], orbicularis orbis muscle [30] or masseters [34,37]). Similar results were obtained in all cases, although Nam et al. [60] reported that NMES on the suprahyoid muscles induced an increase in the anterior hyoid excursion, and that the stimulation of the infrahyoid muscles raised the larynx; thus, they concluded that a combination of these locations should be considered to achieve greater efficacy.

The duration of the sessions varied very little, with most lasting 20 [27,32,33,37,44,47] or 30 min [28,29,30,31,35,36,40,41,42,43]. As was previously described, type II fibres are those which reach fatigue faster, and they are also the ones that activate predominantly with NMES, thus, the excessive duration of the sessions can cause a counterproductive fatigue. However, the duration of the interventions was very diverse, with some being conducted for less than one month [27,28,29,30,31,32,33,34,36,37,38,40,42,43,45,46,47] and others lasting up to two months [35,41]. Regardless of the duration of the intervention, which is also strongly influenced by the cognitive state of the patient and the severity of the DP, two relevant aspects must be highlighted: even the shortest intervention programmes obtained positive effects on the symptoms of the patients (in one of the cases, such positive results were observed only after one week of treatment [47]) and, on the other hand, at least three or four weeks are required to properly evaluate the effects of NMES, as this is the time needed to cause identifiable and significant changes in the muscular physiology through training [61].

The studies that applied NMES as a single treatment technique [28,29,30,35] reported the beneficial effect of this method on DP, as had already been corroborated in a previous meta-analysis [25]. However, the benefits are multiplied when its application is combined with other interventions, such as manual therapy [34,36,37,40,41,42,44,45,47] and conventional swallowing therapy [27,31,32,34,37,38,40,41,42,44,45,46,47]. The greater efficacy of the treatments that combine conventional techniques with NMES had already been observed with longer-lasting effects and shorter intervention times [62,63,64,65]. However, no study combined NMES with other instrumental treatment techniques such as Tonic Tongue (ToTo), which has recently has recently been shown to provide assistance to the performance of isotonic exercises for tongue strength rehabilitation and direct, reliable monitoring through force measurements [66].

Regarding the different variables analysed, one of the most repeated ones was the analysis of the level of oral intake [33,40,41,43,45,46]. This variable improved significantly with all the interventions [40,41,43,46], except for the studies of Mituuti et al. [33], who did not define the impulse time (a parameter that, if chosen incorrectly, can lead to the loss of intervention efficacy), and Guillèn-Sóla et al. [45], who used the sensitivity threshold as the impulse intensity, thus they may have not achieved an effective stimulation of the muscle fibres. It is important to highlight that Sproson et al. [43], in their evaluation of this variable three months after the intervention, observed that the initial improvements achieved were conserved.

The presence and severity of penetration and aspiration was also evaluated [28,33,35,37,41,43,45], obtaining significant improvements in all cases, except in the study that applied NMES at low intensity [33], in which the sessions were excessively long [45], and in the study in which the authors did not use individualised paremeters regarding intensity [37].

The quality of life (a very relevant variable among DP patients [12]) improved in all the studies in which it was evaluated [31,33,34,43], even some time after the end of the intervention [43].

The oral and pharyngeal transit times are very important parameters for the valuation of patients with DP, since these indicate the time that the alimentary bolus takes to reach the upper esophageal sphincter from the oral cavity. These improved with all the analysed interventions [28,29,30,31,33,35,38,47], except, once again, in the study in which NMES was applied up to the sensitivity threshold [33].

Lastly, this review has some limitations that must be pointed out, such as the inclusion of non-controlled and non-randomised studies (which reduces the reliability of the results obtained in them), the small sample size of some of them (which reduces the generalisability and extrapolability of their results) and the exclusion of studies that applied NMES intracranially or invasively. On the other hand, the strengths of this work must also be highlighted, such as delving into the application parameters of a technique that is demonstrated to be essential to reach the maximum efficacy of the physiotherapeutic treatment of DP, as well as the inclusion of a wide range of studies among the obtained results.

Considering the above-mentioned factors, it is recommended to carry out further studies with reliable methodology to establish the most adequate application parameters for NMES and determine the most appropriate combination of techniques to be performed simultaneously and in the same session. This will allow protocolising the use of NMES for the treatment of DP with the aim of reaching the best effects in the shortest time possible.

## 5. Conclusions

NMES has positive effects on the treatment of DP associated with a stroke: it improves the quality of life of the patient, reduces aspirations, restores the capacity to intake solids and reduces the socioeconomic impact of this condition. This technique has beneficial effects as a single treatment, although the attainment of therapeutic objectives is faster when it is combined with active work from the patient, simultaneously, and also when applied as part of a programme that includes other swallowing techniques or exercises.

The application parameters of NMES should be: a frequency of 60–80 Hz, 700 µs of impulse time, an intensity above the motor threshold (respecting the patient’s tolerance) and an application time of 20–30 min, placing the electrodes in the anterior side of the neck. Simultaneously, the patient must be requested to make voluntary contractions of the deficient muscles in order to optimise the increase in muscular strength. Finally, if this technique is part of a treatment that also includes conventional swallowing treatment techniques or strengthening exercises, the treatment objectives can be attained sooner (with treatment durations of four weeks).

## Figures and Tables

**Figure 1 jcm-09-02618-f001:**
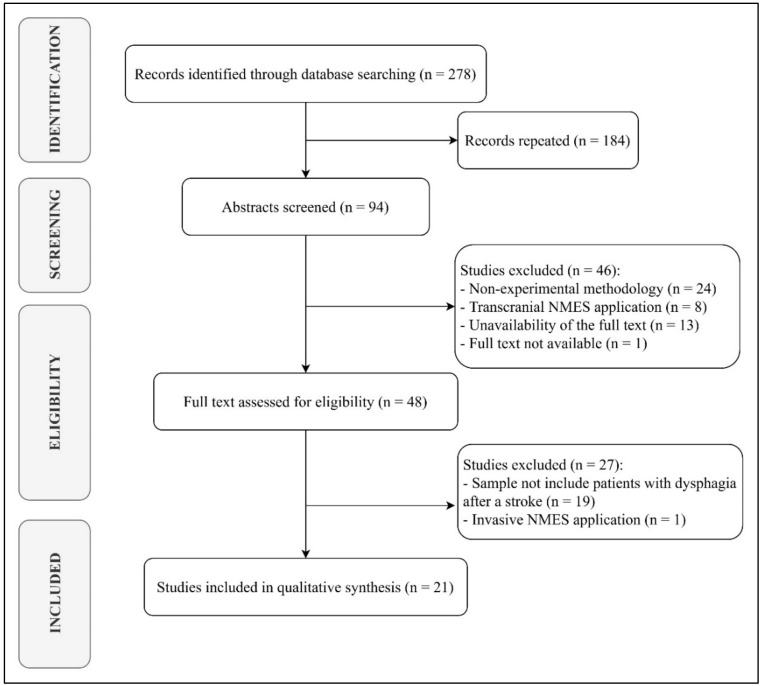
PRISMA chart detailing the article selection process.

**Table 1 jcm-09-02618-t001:** Search strategy according to the focused question (PICO).

Database	Search Equation
PubMed	(“Electric Stimulation Therapy”[Mesh]) AND “Stroke”[Mesh]) AND “Deglutition Disorders”[Mesh]
	(“Stroke”[Mesh]) AND “Electric Stimulation Therapy”[Mesh] AND “Dysphagia”
Medline	(MH “Deglutition Disorders”) AND (MH “Stroke”) AND (MH “Electric Stimulation Therapy”)
	(MH “Stroke”) AND (MH “Electric Stimulation Therapy”) AND “Dysphagia”
Cinahl	(MH “Stroke”) AND (MH “Electric Stimulation”) AND (MH “Deglutition Disorders”)
	(MH “Stroke”) AND (MH “Electric Stimulation”) AND “Dysphagia”
Web of Science	TOPIC: (‘deglutition disorders’) AND TOPIC: (‘stroke’) AND TOPIC: (‘electrical stimulation therapy’)
	TOPIC: (‘dysphagia’) AND TOPIC: (‘stroke’) AND TOPIC: (‘electrical stimulation therapy’)
Scopus	TITLE-ABS-KEY (‘deglutition AND disorders’) AND TITLE-ABS-KEY (‘stroke’) AND TITLE-ABS- KEY (‘electrical AND stimulation AND therapy’)
	TITLE-ABS-KEY (‘dysphagia’) AND TITLE-ABS-KEY (‘stroke’) AND TITLE-ABS-KEY (‘electrical AND stimulation AND therapy’)

**Table 2 jcm-09-02618-t002:** Methodological characteristics of the studies analyzed.

Authors	Design	Sample Size	Inclussion Criteria	Exclussion Criteria	Jadad Scale			
					RD *	BD **	WD ***	Final Score
Bahceci et al. (2017)	ECS	72	Diagnosis of DP in the first 30 days after the stroke. Age between 50 and 75 years.	Diagnosis of cancer, dementia, psychiatric disorder, brain stem disease and/or bleeding (subcortical or bilateral). History of stroke, head and/or neck surgery, impaired swallowing. Smokers	0	2	0	2
Byeon (2016)	ES	47	Diagnosis of DP for more than six months.	Severe cognitive or communicative disorder, depression or nasogastric tube	0	0	0	0
Byeon (2020)	ECS	43	Over 60 years old. Alteration of swallowing after stroke for more than six months of evolution. Korean Mini-Mental State score at least of 20 points.	Receive some treatment for swallowing earlier.	0	1	0	1
Byeon & Koh (2016)	ES	53	Diagnosis of moderate or severe DF for more than six months.	Diagnosis of any mental illness, depression or nasogastric tube	2	1	0	3
Carnaby et al. (2019)	RCCT	53	Stroke for more than two years. Mann Test score less than 18 points.	Previous diagnosis of swallowing disorder. History of head and/or neck surgery	2	2	1	5
Choi (2016)	ES	9	Diagnosis of DP after stroke for less than three months. Mini-Mental State Examination score at least of 24 points.	Pacemaker wearer. Severe communication difficulty (dementia and/or aphasia). Epilepsy. Unstable medical condition. Skin disorders in the head and/or neck.	0	0	0	0
Guillén-Sòla et al. (2017)	RCCT	50	Diagnosis of DP after ischemic stroke. Penetration Aspiration Scale score at least of 3 points.	Diagnosis of previous neurological pathology. Cognitive impairment.	2	1	0	3
Hamada et al. (2016)	RCCT	53	Diagnosis of DP after stroke.	*Not described*	0	0	0	0
Hendy et al. (2019)	RCCT	30	Diagnosis of DP between 1–3 months of evolution. Age between 45–85 years. Conserved cognitive skills	Pacemakers, orthoses and/or metal implants wearer. Pregnancy. Diagnosis of compulsive disorder and/or cancer.	1	0	0	1
Kim et al. (2017)	ES	19	Diagnosis of DP after stroke. Cognitive and the swallowing function preserved	Diagnosis of subarachnoid hemorrhage and/or carotid stenosis. Inability to tolerate NMES.	0	0	0	0
Konecny & Elfmark (2018)	RCCT	108	Diagnosis of DP after stroke. Cooperative patient. Negative water test result.	*Not described*	1	0	0	1
Lee et al. (2019)	ES	40	Diagnosis of DP after stroke.	Diagnosis of previous oral dysfunction and/or stroke. Presence of abnormalities of the oral cavity. Reduced mental capacity and/or severe impairment of cognitive function. Unstable medical condition with inability to swallow.	2	0	0	2
Li et al. (2018)	RCCT	135	Diagnosis of DP. Stroke for more than three months. Age between 50-80 years. Communication skills, movement of the hyoid bone and constriction of the pharynx preserved. Stable health condition.	Diagnosis of progressive stroke, cancer and/or other neurological disorders (amyotrophic lateral sclerosis, multiple sclerosis, Parkinson’s). Radiotherapy treatment. History of head and/or neck surgery. Inability to swallow and/or nasogastric tube.	2	1	1	4
Meng et al. (2018)	RCCT	30	Diagnosis of DP. Stroke for more than six months. Age between 50-80 years.	Pacemaker wearer. Diagnosis of severe pulmonary or cardiac pathology, dementia, aphasia. Non-collaborating patient. Presence of severe aspiration and/or inability to swallow.	2	0	0	2
Mituuti et al. (2018)	ES	10	Diagnosis of DP. Stroke for more than six months. Over 60 years. Regular neurological monitoring. Speech therapy treatment lasting more than six months. Stable oral health condition. Token Test-Short Form score complete.	Diagnosis of cancer. Dental rehabilitation during the intervention period. NMES contraindications (pain and/or intolerance to stimulation).	0	0	0	0
Oh et al. (2017)	ES	8	Diagnosis of DP. Stroke in the last six months. Difficulty closing the lips and to communicate	*Not described*	0	0	0	0
Park et al. (2016)	ES	50	Diagnosis of DP after stroke for more than six months. Ability to swallow against resistance. Cooperator. Mini-Mental State Examination score at least of 24 points.	Diagnosis of psychiatric and/or communication disorder, dementia, aphasia and/or epilepsy. Unstable medical condition. Skin disorders in the head or neck.	2	1	0	3
Park et al. (2019)	ES	10	Diagnosis of DP after stroke. Cough after Water Swallowing Test (3 oz). Initiative to swallow without stimulation of less than six months of evolution.	Pacemaker wearer. Cognitive impairment and/or communication difficulties (dementia and/or aphasia). Unstable medical condition. Skin disorders in the head and/or neck.	0	0	0	0
Simonelli et al. (2018)	RCCT	31	Diagnosis of severe-deep DP between three weeks and three months of evolution after stroke (the first one for the patient). Age between 18–85 years. Stable health condition.	Pacemaker wearer. Cognitive impairment. Diagnosis of epilepsy, depression, cancer and/or neurodegenerative disease. Unstable cardio-pulmonary state. History of head or neck surgery. Previous swallowing treatment.	1	1	0	2
Sproson et al. (2018)	RCCT	30	DP with reduced pharyngeal elevation. Stroke for more than one month. Stable clinical status.	Pacemaker wearer. Diagnosis of serious heart conditions or other neurological pathologies.	2	2	0	4
Zeng et al. (2018)	RCCT	112	Diagnosis of DP after stroke (the first one for the patient). Cooperative patient.	Pacemaker, metal implants and/or orthosis wearer. Critical medical condition. Presence of cognitive impairment. Diagnosis of aphasia, cancer, skin disease, peripheral nerve and/or heart disease, epilepsy. Inability to communicate.	2	1	0	3

ECS: Experimental controlled study. ES: Experimental study. RCCT: Randomized controlled clinical trial. DP: Dysphagia. Jadad scale: * RD: Randomization (1 point if randomization is mentioned; 2 points if the method of randomization is appropriate). ** BD: Blinding (1 point if blinding is mentioned; 2 points if the method of blinding is appropriate). *** WD: Whithdrawals (1 point if the number and reasons in each group are stated). NMES: Neuromuscular electrical stimulation.

**Table 3 jcm-09-02618-t003:** Characteristics of the interventions of the studies analyzed.

Authors	Intervention		Time of Intervention	Number of Sessions (Frequency)	Electrode Position	F	IT	I	
	Experimental Group	Control Group							
Bahceci et al. (2017)	NMES + CT	CT: oral hygiene and dietary modifications, swallowing maneuvers, cranio-cervical postural correction, oral strengthening exercises for lips, tongue and jaw, thermal stimulation with cold, and cognitive, respiratory- and sensory-motor rehabilitation therapies.	4 weeks	20 (5 days/week)	*Not described*	*Not described*	*Not described*	*Not described*	
Byeon (2016)	Group 1: NMES. Group 2: Masako maneuver	---	4 weeks	20 (5 days/week)	Mylohyoid and thyroid muscles	80 Hz	300 ms	*Not described*	
Byeon (2020)	Group 1: NMESGroup 2: NMES + Medelsohn maneuver	Medelsohn maneuver	*Not described*	16 (not described)	Hyoid and cricoid bones	80 Hz	300–700 μs	6.5 mA (increase to painful threshold)	
Byeon and Koh (2016)	Group 1: NMESGroup 2: CT (stimulation of the anterior faucial pillar)	---	3 weeks	15 (5 days/week)	Mylohyoid and thyroid muscles	80 Hz	300 ms	2.5–20 mA	
Carnaby et al. (2019)	NMES + CT	Placebo NMES + CT (swallowing behavior intervention and McNeill therapy)	3 weeks	15 (5 days/week)	Hyoid and cricoid bones	*Not described*	*Not described*	Motor threshold	
Choi (2016)	NMES	---	4 weeks	20 (5 days/week)	Paretic facial area	80 Hz	700 μs	Motor threshold	
Guillén- Sòla et al. (2017)	Group 1: NMES + RMT+ CT Group 2: RMT + CT	CT: educational intervention, oral exercises and compensatory techniques	3 weeks	15 (5 days/week)	Suprahyoid muscles	80 Hz	*Not described*	Motor threshold	
Hamada et al. (2016)	NMES + CT	CT (not described)	*Not described*	*Not described*	Mylohyoid muscle and hyoid bone	80 Hz	700 ms	Sensitive threshold	
Hendy et al. (2019)	NMES + CT	Placebo NMES + CT (thermal stimulation, exercises for strengthening and increasing the motion of the tongue, swallowing exercises, the Medelsohn maneuver, cranial-cervical postural correction and diet modifications)	3 weeks	9 (3 days/week)	Area below the chin and on both sides of the pharynx	80 Hz	300 μs	25 mA (or maximum tolerable without pain)	
Kim et al. (2017)	NMES + CT (swallowing muscle strength training)	---	4 weeks	20 (5 days/week)	Suprahyoid area and sternohyoid muscle	80 Hz	700 μs	5 – 8.5 mA	
Konecny & Elfmark (2018)	NMES + CT	CT: postural correction, respiratory rehabilitation, exercises for the tongue, lips and facial muscles, thermal stimulation and swallowing training	1 week	5 (5 days/week)	Suprahyoid muscles	60 Hz	*Not described*	Motor threshold	
Lee et al. (2019)	NMES + CT (oral stimulation with oral and lingual exercises to train strength and endurance)	---	10 days	20 (2 sessions/day)	Group 1: Masseter and suprahyoid muscles. Group 2: Suprahyoid muscles.	80 Hz	300 ms	7 mA	
Li et al. (2018)	Group 1: NMES Group 2: NMES + CT	CT: changes in dietary habits and postural correction	4 weeks	20 (5 days/week)	Between thyroid and cricroid cartilages and between digastric muscles and hyoid bone	*Not described*	*Not described*	7 mA	
Meng et al. (2018)	NMES + CT	CT: dietary modifications, craniocervical postural correction, swallowing skills training, Shaker and Medelsohn maneuvers, esophageal balloon dilation and respiratory exercises.	2 weeks	10 (5 days/week)	Group 1: suprahyoid muscles and cranial and distal to the thyroid cartilage. Group 2: genohyoid and mylohyoid muscles	80 Hz	---	25 mA or motor threshold	
Mituuti et al. (2018)	NMES	---	4 weeks	12 (3 days/week)	Mylohyoid and thyroid muscles	80 Hz	700 μs	Sensitive threshold	
Oh et al. (2017)	NMES	---	4 weeks	20 (5 days/week)	Oral orbicular muscle	80 Hz	700 μs	Motor threshold	
Park et al. (2016)	NMES	---	6 weeks	30 (5 days/week)	Sternohyoid muscles	80 Hz	700 μs	Group 1: sensitive threshold. Group 2: motor threshold	
Park et al. (2019)	NMES	---	4 weeks	20 (5 days/week)	Below the chin and thyroid cartilage	80 Hz	700 μs	25 mA or motor threshold	
Simonelli et al. (2018)	NMES + CT	CT: lingual, oral, facial and pharyngeal exercises, laryngeal elevation exercises, Medelsohn and Masako maneuvers, Shaker exercises and thermal stimulation	8 weeks	40 (5 days/week)	Thyroid muscles	80 Hz	300 μs	7.8–12.5 mA	
Sproson et al. (2018)	NMES + CT	CT: dietary modifications and three swallowing strengthening exercises.	4 weeks	20 (5 days/week)	Sternohyoid muscles	30 Hz	---	Minimum motor threshold	
Zeng et al. (2018)	NMES + CT + PT	PT (platelet inhibitors, hypolipimics, antihypertensives, euglycemics and facilitators of microcirculation) + CT (massage on cheeks, tongue, retropharyngeal wall, pharyngeal-palatal area and lips with cotton soaked in ice)	24 days	24 (1 per day)	Hyoid bone and thyroid cartilage	---	800 ms	28 mA	

NMES: Neuromuscular electrical stimulation. CT: Conventional therapy. F: Frequency. IT: Impulse time. I: Intensity. ---: not applicable.

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
