# Peer review of "Effectiveness of Different Application Parameters of Neuromuscular Electrical Stimulation for the Treatment of Dysphagia after a Stroke: A Systematic Review"

_jcm, 2020, doi:10.3390/jcm9082618_

Round 1
Reviewer 1 Report
wel done, the paper is now more readable and clear
Author Response
Dear Editor and Reviewers of the Journal of Clinical Medicine:
Thank you very much for your suggestions and contributions to improve the quality of the manuscript. Following your indications, we respond, point by point, to the reviewers' comments.
- Wel done, the paper is now more readable and clear.
The authors acknowledge the positive recognition of our work.
Once again, thank you very much for the time spent and the interest shown in this work; as well as in the positive evaluations you have given of it.
Receive a warm greeting,
The authors.
Reviewer 2 Report
The authors presented an interesting review on the effectiveness of neuromuscular electrical stimulation for treating disphagic subjects.
The article provides an exhaustive overview of the topic. It is well written and easy to read.
I have a few minor concerns:
-) remove the red color throughout the text
-) remove the "..." at line 67
-) I would add a Picture showing how/where the electrodes are placed
-) I would suggest to move the Tables reported in the Supplementary Section to the main manuscript. Personally, I believe that a reader would be more interested in having a summary of the procedures rather than seeing Tables about the methodology (that can be moved, instead, to the Supplementary Section).
Author Response
Dear Editor and Reviewers of the Journal of Clinical Medicine:
Thank you very much for your suggestions and contributions to improve the quality of the manuscript. Following your indications, we respond, point by point, to the reviewers' comments.
- Remove the red color throughout the text and the "..." at line 67
The authors use the color red in the text to highlight the changes made after the first corrections made by previous reviewers.
The red color has been removed and ellipsis of the indicated line also.
- I would add a picture showing how/where the electrodes are placed.
The authors do not have possession of any photo or drawing with the characteristics requested.
At the same time, we hope that the detailed description that we have made on the suitable location of the electrodes is sufficient.
- I would suggest to move the Tables reported in the Supplementary Section to the main manuscript. Personally, I believe that a reader would be more interested in having a summary of the procedures rather than seeing Tables about the methodology (that can be moved, instead, to the Supplementary Section).
The table previously included in the Supplementary Section has been moved to the main manuscript (it is now Table 3).
Once again, thank you very much for the time spent and the interest shown in this work; as well as in the positive evaluations you have given of it.
Receive a warm greeting,
The authors.
This manuscript is a resubmission of an earlier submission. The following is a list of the peer review reports and author responses from that submission.
Round 1
Reviewer 1 Report
The paper entitled "Efficacy of neuromuscular electrical stimulation for the treatment of dysphagia after a stroke: a systematic review" is a work aiming to focus on the reports about NMES responsiveness to dysphagia stroke-related.
Introduction is well written and adequate in length and in content.
Methods are too vague and not adequately described:
- Fig1 shows the PRISMA chart, but in "METHODS" the authors did not mention it. I suggest to rewrite in details in methods (if) they adopted the PRISMA statement for data extraction, as well the jadad scale for assessing the quality of the studies, if the systematic review was registered on Prospero and so on.
- on the opposite, results and their tables are well done, clear and exhaustive, but, too long. Being a lots of quantitative data, a meta analysis
- In tables, I suggest to report the acronym of JADAD in the table legend.
- discussion is too long and partially repeated data reported in "results".
English style is fluent and well readable.
In conclusion, I suggest to:
- rewrite a more detailed M&M section
- shorten the results and provide a meta analysis of the whole data
- rewrite the conclusion, avoiding to repeat results already reported .
Author Response
Dear Editor and Reviewer of Journal of Clinical Medicine:
Thank you very much for your suggestions and contributions to improve the quality of the manuscript. Following your indications, we respond, point by point, to your comments.
In the text, all the modified or added sentences have been written in red to facilitate the correction by the reviewers.
- Methods are too vague and not adequately described: Fig. 1 shows the PRISMA chart, but in "METHODS" the authors did not mention it.
The reference to Figure 1 appears on page 2 (line 70).
- I suggest to rewrite in details in methods (if) they adopted the PRISMA statement for data extraction, as well the jadad scale for assessing the quality of the studies, if the systematic review was registered on Prospero and so on.
The authors have expanded the section of material and methods as you indicate.
- On the opposite, results and their tables are well done, clear and exhaustive, but, too long. Being a lots of quantitative data, a meta analysis
Thank you very much for the positive evaluation you make of the Results section.
Conducting a meta-analysis it was considered but it was not feasible due to the diversity in the assessment tools used by the research analyzed.
- In tables, I suggest to report the acronym of JADAD in the table legend.
Jadad is not an acronym. This scale has this name by its author: Alejandro Jadad Bechara.
- Discussion is too long and partially repeated data reported in "Results".
The authors have omitted some Discussion phrases that repeated information from the Results section.
- In conclusion, I suggest to: rewrite the conclusion, avoiding to repeat results already reported.
The authors have avoided repeating information in the Conclusions section: we removed sentences from the Discussion and have included them in Conclusions in order to emphasize the findings.
Once again, thank you very much for the time spent and the interest shown in this work; as well as in the positive evaluations you have given of it.
Receive a warm greeting,
The authors.
Reviewer 2 Report
Although the subject of this paper is interesting the review is unsatisfactory. The efficacy of neuromuscular electrical stimulation for the treatment of dysphagia has been reviewed by several excellent papers. The authors of this manuscript have not discussed those papers and have not given any satisfactory reasons of writing their review manuscript. In addition, there are numerous misunderstanding and errors. For example, the authors state that physiotherapeutic method is the only ones capable of restoring physiological swallowing which is misleading. Furthermore, the authors use papers of last five years only. This manuscript lacks of the scientific strength compatible with publication in Journal of Clinical Medicine.
Author Response
Dear Editor and Reviewer of Journal of Clinical Medicine:
Thank you very much for your suggestions and contributions to improve the quality of the manuscript. Following your indications, we respond, point by point, to your comments.
In the text, all the modified or added sentences have been written in red to facilitate the correction by the reviewers.
- Although the subject of this paper is interesting the review is unsatisfactory. The efficacy of neuromuscular electrical stimulation for the treatment of dysphagia has been reviewed by several excellent papers. In addition, there are numerous misunderstanding and errors. For example, the authors state that physiotherapeutic method is the only ones capable of restoring physiological swallowing which is misleading. Furthermore, the authors use papers of last five years only. This manuscript lacks of the scientific strength compatible with publication in Journal of Clinical Medicine.
The authors have corrected the manuscript according to their advice and those of the other reviewers.
Unlike previous bibliographic reviews, note that this review has only included the application of surface NMES (applying its invasive or transcranial application as the exclusion criteria).
The authors hope that this new version of the manuscript is to your liking.
Receive a warm greeting,
The authors.
Reviewer 3 Report
The authors presented an interesting review on the efficacy of neuromuscular electrical stimulation to recover dysphagic patients after stroke.
The manuscript is well written and easy to read. I would suggest the following minor revisions:
-) include at least one picture to better describe how neuromuscular electrical stimulation is performed.
-) include a relevant recent study on treating dysphagia through mechanical exercises aimed at improving the performance of the tongue musculature:
Milazzo, M., Panepinto, A., Sabatini, A. M., & Danti, S. (2019). Tongue Rehabilitation Device for Dysphagic Patients. Sensors, 19(21), 4657.
-) the authors excluded publications in a language that was not English or Spanish. Is there a specific reason why Spanish studies were taken into account instead of, for instance, French or Italian works?
-) include quantitative results. In many cases, the authors stated: "results were positive" (e.g., line 141). My questions are: Which are the benchmarks to assess the improvement? How much these benchmarks showed variations? Are the differences statistically relevant? Please revise the manuscript including more quantitative details (if any).
Author Response
Dear Editor and Reviewer of the Journal of Clinical Medicine:
Thank you very much for your suggestions and contributions to improve the quality of the manuscript. Following your indications, we respond, point by point, to your comments.
In the text, all the modified or added sentences have been written in red to facilitate the correction by the reviewers.
- Include a relevant recent study on treating dysphagia through mechanical exercises aimed at improving the performance of the tongue musculature: Milazzo, M., Panepinto, A., Sabatini, A. M., & Danti, S. (2019). Tongue Rehabilitation Device for Dysphagic Patients. Sensors, 19(21), 4657.
In the Discussion section, the authors have made reference to the study device in the research that you recommend and has been adequately cited.
- The authors excluded publications in a language that was not English or Spanish. Is there a specific reason why Spanish studies were taken into account instead of, for instance, French or Italian works?
That exclusion criterion has been removed. Actually, no article was excluded due to its language. Only one article was rejected due to not being able to access the full text.
The authors have corrected the exclusion criteria and Figure 1.
- In many cases, the authors stated: "results were positive" (e.g., line 141). My questions are: Which are the benchmarks to assess the improvement? How much these benchmarks showed variations? Are the differences statistically relevant? Please revise the manuscript including more quantitative details (if any).
In the three sentences in which we use that phrase we really want to refer to the fact that there were statistically significant changes.
We have replaced the three sentences with expressions that clearly express the results obtained.
Once again, thank you very much for the time spent and the interest shown in this work; as well as in the positive evaluations you have given of it.
Receive a warm greeting,
The authors.
Round 2
Reviewer 1 Report
Dear Authors,
the revised paper has been noticeably improved and it is now ready to be acceptable in the present form.
Best wishes.
Reviewer 2 Report
The authors have submitted a revised version of the manuscript. From my point of view, I am very sorry to say that the revised manuscript is not satisfactory and lacks of the scientific strength compatible with publication in Journal of Clinical Medicine.
Comments
- The efficacy of neuromuscular electrical stimulation for the treatment of dysphagia has been reviewed by several excellent papers. This manuscript has not provided any critical discussion about those and even they have not cited those papers.
Examples of previous review papers:
a. Chen, Y.W.; Chang, K.H.; Chen, H.C.; Liang, W.M.; Wang, Y.H.; Lin, Y.N. The effects of surface neuromuscular electrical stimulation on post-stroke dysphagia: A systemic review and meta-analysis. Clin. Rehabil. 2016, 30, 24–35.
b. Carnaby-Mann, G.D.; Crary, M.A. Examining the evidence on neuromuscular electrical stimulation for swallowing: A meta-analysis. Arch. Otolaryngol. - Head Neck Surg. 2007, 133, 564–571, doi:10.1001/archotol.133.6.564.
2. Although the title of this paper is ‘’Efficacy of neuromuscular electrical stimulation for the treatment of dysphagia after a stroke: a systematic review”---there is no critical discussion about the efficacy of this in treating dysphagia. Rather the authors more concentrate on methods of neuromuscular electrical stimulation.
3. This review paper even does not give definition of neuromuscular electrical stimulation. Neuromuscular electrical stimulation has been investigated for dysphagia at sensory and motor intensities. This paper does not give any clear indication about these.
4. Previous systemic review and meta-analysis on neuromuscular electrical stimulation for dysphagia suggested that the evidences regarding its efficacy for treating dysphagia is still controversial/limited. However, the authors of this review paper stated that “…an physiotherapeutic methods (the only ones capable of restoring physiological swallowing)”. The reference that is used for this statement even not deals with the efficacy of physiotherapeutic methods in dysphagia management.
5. This manuscript does not give proper introduction of other promising strategies (e.g.; non-invasive brain stimulation, pharyngeal electrical stimulation, other peripheral stimulation, recent pharmacological strategies like activation of TRP channels etc.).
6. There is no critical discussion comparing the efficacy of on neuromuscular electrical stimulation with other promising strategies for dysphagia treatment. Rather the authors more discus on the methodology of neuromuscular electrical stimulation which is not the aim of this review.